# Secretomic Insights into the Pathophysiology of *Venturia inaequalis*: The Causative Agent of Scab, a Devastating Apple Tree Disease

**DOI:** 10.3390/pathogens12010066

**Published:** 2022-12-31

**Authors:** Yash Paul Khajuria, Bashir Akhlaq Akhoon, Sanjana Kaul, Manoj Kumar Dhar

**Affiliations:** School of Biotechnology, University of Jammu, Jammu 180006, India

**Keywords:** *Venturia inaequalis*, transposable elements, genome annotation, secreted proteins, virulence

## Abstract

Apple scab, caused by *Venturia inaequalis*, is one of the world’s most commercially significant apple diseases. The fungi have a catastrophic impact on apples, causing considerable losses in fruit quality and productivity in many apple-growing locations despite numerous control agents. Fungi secrete various effectors and other virulence-associated proteins that suppress or alter the host’s immune system, and several such proteins were discovered in this work. Using state-of-the-art bioinformatics techniques, we examined the *V. inaequalis* reference genome (EU-B04), resulting in the identification of 647 secreted proteins, of which 328 were classified as small secreted proteins (SSPs), with 76.52% of SSPs identified as anticipated effector proteins. The more prevalent CAZyme proteins were the enzymes engaged in plant cell wall disintegration (targeting pectin and xylanase), adhesion and penetration (Cutinases/acetyl xylan esterase), and reactive oxygen species formation (multicopper oxidases). Furthermore, members of the S9 prolyl oligopeptidase family were identified as the most abundant host defense peptidases. Several known effector proteins were discovered to be expressed during the *V. inaequalis* infection process on apple leaves. The present study provides valuable data that can be used to develop new strategies for controlling apple scab.

## 1. Introduction

Apple scab by *Venturia inaequalis* Cooke (Wint.) is one of the most devastating diseases of apples. The fungus infects the leaves, fruits, branches, petioles, and stems, but leaves and fruits are the most affected parts of the plant [1]. In severe cases of infection, production losses of more than 70% have been reported. Temperate regions with humid climates are highly susceptible to this disease [2]. Different types of fungicides are used 16–20 times to control the disease [3]. This raises environmental and human health concerns and also increases overall production costs. Scab-resistant apple cultivars have been introduced as an alternative approach to disease control [4]; however, the disease has re-emerged in areas where resistant cultivars have been cultivated due to resistance breakdown by the fungus [5,6]. Therefore, new strategies based on the molecular mechanisms of host–pathogen interactions are needed to seek holistic solutions to combat disease.

*V. inaequalis* is a semi-biotropic fungus that survives the winter as pseudothecia on rotting leaves [7]. Crossover occurs between opposite mating types and ascospores are produced in the pseudothecia. In early spring, ascospores mature and are released by the pseudothecia, which infect young leaves and shoots. Further infection is initiated by conidia, giving rise to many secondary infection cycles [1]. The fungus does not enter cells but is confined to subcutaneous areas without significant damage to the host tissues [8]. Several studies have reported very high genetic diversity within different populations of *V. inaequalis* [9,10]. High genetic diversity has been attributed to sexual reproduction, which allows genetic recombination.

Plants have a sophisticated innate immune system to keep pathogens away [11]. However, pathogens secrete a wide variety of secretory proteins (effectors) to suppress or alter the host’s immune system [12]. These effectors are small cysteine-rich secreted proteins that help pathogens colonize host plants. They can help pathogens obtain nutrients from the host. They can also aid fungi to hide from the host’s immune system by sequestering pathogen molecules. Plants can recognize molecules of non-native microbial origin, called pathogen-associated molecular patterns (PAMPs), which in turn activate a plant’s defense mechanism known as pattern-triggered immunity (PTI) [13]. Understanding the localization of candidate effectors in host tissues may suggest their mechanism of action. Apoplast effectors inhibit various types of lytic enzymes that degrade fungal components and prevent PTI [14]. Some effectors translocate to cell membranes and other compartments where they target different types of immune components. Components attacked include catalase, H_2_O_2_ homostat, protein stability, ubiquitin proteasome, sRNA silencing machinery, and plant hormones [15]. However, plants also have sophisticated mechanisms to counteract these effectors and other proteins associated with virulence. Plant-derived disease resistance (R) gene products can recognize some of these effectors, thus conferring effector-triggered immunity (ETI) in the host. Many of these interactions have been identified in the *Venturia* apple pathosystem [16].

A number of virulence-associated proteins have been identified and characterized in *V. inaequalis*. Changes in phytohormone levels have been noted during the infection process [17]. Interestingly, several genes overexpressed during the growth on cellophane have been reported in *V. inaequalis* [18,19]. Advances in genomics and transcriptomics has made it possible to sequence and analyze the genomes and transcriptomes of several fungal pathogens including *V. inaequalis,* leading to the identification of many secreted proteins associated with virulence [20,21]. The *Venturia* genome has been sequenced and annotated from different isolates in varied geographic regions of the world [22,23,24,25]. Deng and coworkers annotated the *V. inaequalis* and *V. pirina* genomes and predicted several effectors associated with pathogenesis [23]. Le Cam et al. sequenced 87 isolates from different populations of *Venturia* and also reported the genome sequence of *V. inaequalis* reference genome (EU-B04) using PacBio technology [25]. In the present study, the EU-B04 reference genome was further analyzed using updated bioinformatics tools to predict and annotate its several secretome proteins. Transcriptome analysis was also performed to identify differentially expressed genes (DEGs) in *V. inaequalis* during the apple leaf infection process. Therefore, the present study will help us to better understand the mechanisms implemented by *V. inaequalis* during apple colonization.

## 2. Materials and Methods

### 2.1. Repeat Annotation

We used RepeatModeler2 [26] with default parameters to annotate the transposable elements of the EU-B04 genome assembly. For the identification of long terminal repeat (LTR) retrotransposons, we employed the LTR pipeline, an optional module of RepeatModeler2. The detected repeat elements were then utilized to soft mask the genome assembly with RepeatMasker v4.1.2 [27].

### 2.2. Gene Prediction

Gene models were inferred using BRAKER2 [28]. The initial gene sets were predicted using GeneMark-ES v4.10, and then the intron hints were generated by the ProtHint pipeline, using protein data from the fungal OrthoDB database (https://v100.orthodb.org/download/odb10_fungi_fasta.tar.gz, accessed on 16 February 2022) as a resource of benchmark protein sequences to train GeneMark-EP+ [29]. Finally, the GeneMark-EP+ predictions, along with the intron hints generated from the RNA-Seq data obtained from the NCBI’s Sequence Read Archive via VARUS using *Venturia inaequalis* as a binomial name, were utilized to train Augustus to produce the final gene models. BUSCO was then used to check the completeness of predicted gene models using the dataset ascomycota_odb10 (10 September 2020).

### 2.3. Secretome Analysis

A well-known protein sorting signal, the secretory signal peptide, was discovered using a combination of SignalP-6.0 [30] and Phobius [31] tools. SignalP cannot detect transmembrane proteins, so we used TMHMM-2.0 [32] for this purpose. Final signal peptide protein selection was achieved using SignalP-TMHMM and Phobius Consensus predictions. Indeed, not all proteins are secreted via the ‘conventional’ signal peptide-dependent pathway, but a number of eukaryotic proteins are secreted via atypical ‘non-conventional’ signal peptide-independent mechanisms. Therefore, we used SecretomeP [33] and OutCyte [34] to predict non-classical secreted proteins. To reduce the possibility of false predictions, we chose to include only proteins predicted to be non-classically secreted by both algorithms (SecretomeP and OutCyte).

To predict subcellular protein localization, the BUSCA program was used [35]. The program combines a number of computational tools such as DeepSig, TPpred3 predictor, PredGPI, ENSEMBLE3.0, and BetAware, and uses machine/deep learning algorithms to detect and predict properties important for protein subcellular distribution. Several studies found that CELLO [36] and DeepLoc [37] are generally more effective than SecretomeP in predicting non-classical secreted proteins of the extracellular category [38]. Therefore, we chose to include them in this study as well, and only the consensus hits predicted as ‘extracellular’ by all three methods (BUSCA, CELLO, and DeepLoc) were considered for further analysis. Glycosylphosphatidylinositol (GPI) anchors are known to bind secreted proteins to the fungal cell surface, hence NetGPI [39] was used to exclude such proteins from the discovered *V. inaequalis* secretome.

### 2.4. Identification of Candidate Effectors

EffectorP version 3.0 [40], a machine learning approach, was used for effector prediction. In addition, proteins that resembled effectors reported in the Pathogen-Host Interaction Database (PHI-base) were also considered effectors. The combined findings of the two approaches resulted in the final collection of putative effectors. We used the ApoplastP program [41] to predict the host cell localization of effectors identified by PHI-base screening.

### 2.5. Functional Annotation

Gene annotation was performed with default settings using Funannotate v1.8.10. Annotation of functional domains was conducted using EggNOG-mapper v5, InterProScan v5, and HMMer scans against the Pfam-A database. Carbohydrate-activating enzymes (CAZymes) and peptidases were identified using the dbCAN CAZYmes database and the MEROPS protease database. BlastP searches were also performed against the Uniport/Swissport and NCBI non-redundant (NR) and Refseq databases.

### 2.6. Preparation of Samples

Isolates of *V. inaequalis* were obtained from the Kashmir Valley cultivar Red Delicious. Single spore isolation was carried out, and cultures were grown on a potato dextrose agar (PDA) medium. Mycelium harvested from PDA plates was immediately stored in liquid nitrogen and stored at −80 °C until further use. Mature leaves were cut from the apple plant, surface sterilized and placed in the correct orientation on moist paper in a Petri dish. *V. inaequalis* mycelium was washed with distilled water and placed onto cut apple leaves [42]. After 5 days, the fungus was harvested from the leaves.

### 2.7. RNA Isolation and Library Preparation

Total RNA isolation was performed using the modified Trizol method [43]. Quality was checked using RIN > 8 on a Bioanalyzer (Agilent, Santa Clara, CA, USA; 2100). A total of 1 µg of high-quality RNA was used for library preparation using the Illumina TruSeq RNA Sample Prep Kit v2. After first-strand and second-strand cDNA synthesis, the ends were repaired and adenylated [44]. Library quantification was performed using the Qubit™ RNA Assay Kit for the Qubit 2.0^®^ Fluorometer. Next-generation transcriptome sequencing was performed on the Illumina HiSeq 2500 platform.

### 2.8. Transcriptome Profiling

We used the Fastp tool [45] to remove low-quality sequencing adapters and read bases from the raw sequencing data. For quality control, measurements were examined using FastQC. To align the reads with the *V. inaequalis* genome, we used the genomic sequence of the EU-B04 genome and the required GTF/GFF from the BRAKER2 study. Proper indices and FASTQ read alignments were generated using STAR (v2.6.1b), and the resulting SAM files were sorted by coordinate using samtools (v1.14) and exported to BAM format for further processing. A count matrix was generated with HTSeq software (v1.99.2) and the resulting counting data was used for differential expression analysis. The trimmed mean method was used for statistical modelling of the count data using the R computing environment (4.1.2) and the Bioconductor package DESeq2 (v1.20).

## 3. Results and Discussion

The *V. inaequalis* genome is expected to be around 100 MB in size [46]. Various researchers have reported *V. inaequalis* assembly sizes ranging from 40 Mb to 72.3 Mb [22,23,24]. Le Cam et al. sequenced the EU-B04 reference genome, which is 72.79 Mb in size [25].

### 3.1. V. inaequalis Transposable Elements Represent Half of the Overall Genome Size

*V. inaequalis* has received substantial research as a model for infections on woody hosts. There are several genome sequences available for *V. inaequalis*, and researchers determined that pathogenic strains of this model have a higher repeat content than other species, with repeat regions occupying up to 47% of their genome sequence [23,24].

We employed RepeatModeler2 to annotate transposable elements (TE) in the EU-B04 genome assembly. The approach overcomes several drawbacks of the previous RepeatModeler method, notably the lack of a comprehensive, nonredundant library of full-length consensus sequences [26]. We also utilized its built-in optional LTR module to find LTR elements in the genome. A total of 48.83% of the EU-B04 genome was found to consist of various types of repetitive elements, of which 25.44% were class I LTR elements, mainly Gypsy/DIRS1 (15.52%) and Ty1/Copia (8.83%) superfamilies. Class II elements, accounting for 4.68% of all detected repeats, were mostly from the hobo-Activator superfamily (3.96%). Transposable elements of unknown type were found to account for 17.93% of the identified transposable elements. Figure 1 gives a detailed description of the repeats detected in the EU-B04 genome. Overall, the findings showed that only 51.17% of *V. inaequalis* genome belongs to a non-repetitive region containing all biological genes and other regulatory sequences.

### 3.2. Annotated Genome of V. inaequalis (EU-B04) with a BUSCO Completeness Score of 99.1%

According to the published reports on *V. inaequalis*, the number of gene models in *V. inaequalis* isolates ranges from 11,600 to 13,761 [23,24,25]. Using the BRAKER1 pipeline, Le Cam et al. [25] identified around 11,600 genes in EU-B04. The authors used AUGUSTUS version 3.2.3 to predict the genes, which were backed by extrinsic hints obtained by their in-house RNA studies. Using the updated version of AUGUSTUS 3.3.3 supported by hints from the current fungal OrthoDB database and available RNA-seq data for *V. inaequalis*, we were able to predict 1363 additional genes. In total, the BRAKER2 pipeline identified 12,970 gene models in the EU-B04 genome. The quality of the gene prediction was evaluated by comparing universal single-copy orthologs (BUSCO) with the available Ascomycota dataset. According to the BUSCO results, the predicted genetic model represents most of the genome, with a BUSCO completeness score of 99.1% (86.1% single-copy complete genes and 13.0% duplicated complete genes), with only 0.2% fragmented and 0.7% missing genes, while Le Cam et al. reported 98.47% completeness in their study [25].

### 3.3. V. inaequalis Secretome Accounts for Approximately 5% of Its Total Proteome Size

When the entire *V. inaequalis* proteome (12,967) was examined for secretory proteins, SignalP-6.0 results showed that a total of 1334 proteins contain signal peptides. Among them, TMHMM2.0 identified 245 proteins as having transmembrane signaling, while the remaining 1089 proteins were chosen for further investigation. The Phobius program which predicts both signal peptides and transmembrane topology discovered 1503 non-transmembrane classical secreted proteins in *V. inaequalis*. Finally, the 1046 non-transmembrane signaling proteins identified as common predictors by SignalP/TMHMM and Phobius were selected and considered to be released via the classical secretory pathway. To predict unconventionally secreted proteins, we employed the SecretomeP and OutCyte prediction algorithms. Unlike SecretomeP, which utilizes conventional proteins by eliminating signal peptides, the OutCyte module (OutCyte-UPS) uses an experimental secretome data source to detect potential unconventional protein secretions (UPS) using the characteristics obtained directly from protein sequences. Outcyte detected 1233 proteins secreted via non-canonical pathways, while SecretomeP identified 5723 proteins. Both SecretomeP and Outcyte predicted 549 proteins as common non-classical secreted proteins. After combining classical (1046) and non-classical (549) proteins, a total of 1595 proteins were predicted as secreted proteins. Deng and colleagues [23] found 1131 to 1622 secreted proteins in different *V. inaequalis* isolates. Despite the fact that the bioinformatics tools used in this work are more in number and newer (e.g., SignalP 4.0 compared to SignalP 6.0) (Figure 2), they yielded approximately the same number (1595) of secreted proteins. However, our approach was totally different as Deng et al. chose to combine all predictions from the selected software to avoid a false negative exclusion, while we chose to include only common predictions from many software programs to avoid false positives. In previous studies, CELLO and DeepLoc were better predictors of non-classical secreted proteins of the extracellular class than SecretomeP [38]. According to researchers, DeepLoc outperforms other programs in predicting proteins localized to extracellular compartments in fungal datasets [35]. Therefore, we decided to include these programs in our study as well to predict the subcellular localization of the predicted secreted proteins. Deeploc, BUSCA, and Cello found 946, 939, and 759 extracellular proteins, respectively, and 664 were common predictions of all three programs. The final secretome of *V. inaequalis* (647 predicted proteins) was obtained by filtering GPI-anchored proteins (17) that facilitate protein binding to the outer face of the lipid bilayer [47].

### 3.4. Three-Quarters of V. inaequalis Small Secreted Proteins (76.52%) Are Effectors That May Contribute to Its Virulence

In the secretome, several small secreted proteins (SSPs) are known to act as virulence factors involved in disease development or as avirulent factors that induce resistance (R) gene-mediated protection. SSPs were defined as secreted proteins with a maximum length of 300 amino acids. A total of 328 proteins representing 50.69% of the secretome were identified as SSPs. Researchers have identified several potential small secreted proteins known as effectors that contribute to virulence. These effectors develop a wide range of plant microbial associations, ranging from beneficial to harmful. A total of 225 proteins (68.59%) from SSP were recognized as effectors by EffectorP 3.0. Of these, 39 showed homology to known effectors available in the PHI-base. In addition, 26 SSPs identified as non-effectors by EffectorP showed homology to known effectors in PHI-base. These 26 proteins were added to the final effector list recognized by EffectorP, bringing the number of effectors to 251. Localization analysis showed that 176 effectors (70.11% of all effectors) are apoplastic in nature, although 18.18% of them could also target the host cytoplasm (dual-target effectors). The longest secretome protein was approximately 918 amino acids (aa) and the smallest was 67 aa. The shortest protein lengths of SSP and effector were also the same (67 aa), while the longest protein lengths of SSP and effector were 300 and 299 aa. The average length was 294, 174, and 165 aa in the secretome, SSP, and effectome, respectively (Figure 3).

### 3.5. Functional Annotations of V. inaequalis Secretome and Effectome 

Secretomics research sheds light on the diverse CAZyme weapons used by microorganisms during biomass degradation. In the *V. inaequalis* genome, GH predominated (222), followed by AA (95) and GT (94) in approximately equal numbers. CE was the third-abundant class (88) in CAZymes. PL and CBM also showed minimal presence (13, 3). GH predominated in the secretome (79) as well, followed by AA (46) and CE (33). The GH family plays an important role in pathogen infection and host colonization. Secretome analysis of *V. inaequalis* revealed the presence of several GH families (Figure 4) and the most common GH families were GH43 (15.18%) and GH28 (12.65%). 

The GH43 family included known enzymes with β-xylosidase, α-L-arabinofuranosidase, arabinanase, and xylanase activities, whereas the GH28 family included polygalacturonase and rhamnogalacturonase. The enzymes of the GH28 family promote host colonization [48]. GH28 polygalacturonases have been reported to be involved in plant tissue maceration and soft rot, and the presence of this family has also been confirmed in *Monilinia* species, which causes brown rot [49]. A GH3 family protein containing a fibronectin type III-like domain was the third most abundant (8.86%) GH enzyme. In addition, other GHs such as cellulase (GH5), maltase-glucoamylase (GH31), β-galactosidase (GH35), GH128 with PAN domain, and phosphotyrosine phosphatase (GH131) were slightly more abundant than other GHs. Secretome analysis revealed that cutinases (CE5) were the most common enzyme family (11.37%) among the identified CEs. CE5 is involved in the enzymatic hydrolysis of cutin and is commonly released by plant pathogens, allowing them to cross the cuticle [50]. AA1 dominated (30.43%) the AAs identified in the *V. inaequalis* secretome, and it was also the second most abundant enzyme family after CE5. Peroxidase (AA2), GMC oxidoreductase (AA3), and AA9 (endoglucanase activity) showed similar abundance (15.21%), although three GMC oxidoreductases with combined AA3 and AA8 domains were also present in the secretome. Secretome analysis also revealed the presence of PL genes such as PL1(2), PL3(1), and PL4(4).

Peptidases are known to play a role as virulence factors in fungal infections, and their participation in pathogenicity has been described in various fungal pathogens [51]. In the present study, a significant number of serine peptidases (79.54%) were observed in the *V. inaequalis* secretome. Furthermore, among the identified peptidases, 9.09% were asparagine peptidases, 6.81% were metallopeptidases, and 2.27% were threonine peptidases. In addition, a single copy of a serine carboxypeptidase Y inhibitor (phosphatidylethanolamine-binding protein) was also identified. Deng and coworkers [23] also reported a similar number of peptidases (31–38) from various *V. inaequalis* isolates. Carboxylesterase (S09X) was the most prominent protease in the genome and secretome of *V. inaequalis*. S33 and C19, the second and third most abundant peptidases in the *V. inaequalis* genome showed no dominance in the secretome. C19 was completely absent from the secretome, but there was one copy of the alpha/beta hydrolase enzyme from the S33 family. A subtilase family (S08A), a serine carboxypeptidase (S10), and an endopeptidase with a prokumamolisin activation domain (S53) were other predominant peptidases identified in the *V. inaequalis* secretome. In addition, small numbers of gamma-glutamyl transpeptidase and aspartyl protease were also detected.

Of the predicted effectors, functional annotation was available for only 23.90% of effector proteins. Functional characterization of the *V. inaequalis* effectome revealed the presence of 21 CAZymes. Of these, 42.85% were CE, 23.80% AA, and 19.04% GH. We also observed the presence of one PL (PL3) and two CBMs: CBM50 (LysM domain) and CBM63 (Lytic transglycolase). The LysM protein has been found to scavenge chitin oligosaccharides from fungal pathogens and evade host defense responses [52]. Of the CEs, 77.77% were cutinases/acetyl xylan esterases belonging to the CE5 family, and the remaining CEs were from the CE1 (prolyl oligopeptidase family/esterase-PHB depolymerase) and CE12 (GDSL-like lipase/acylhydrolase) families. The GH11 family (concanavalin A-like lectins/glucanases) contributed to 50% of the identified GHs and the remaining 50% were from the GH131 (β-glucanase) family. Concanavalin A (ConA) is a lectin (carbohydrate-binding protein) that binds complex carbohydrates and contains lectin-like peptidase A4 (PF01828). ConA was found to inhibit the germination of *S. scitamineum*, a fungal pathogen that causes smut in sugarcane [53,54]. It binds to cell surface mannose residues and the incubation of fungal cells with Con A reduces adhesion, hence *V. inaequalis* may be releasing these virulence factors to counteract it. Members of the GH11 family have also been identified as virulence factors in *Botrytis cinerea*, and its mutants have shown reduced virulence [55]. The GH131 enzyme is commonly expressed during colonization of plant tissues by fungal pathogens [56]. Furthermore, we found that all effector AAs detected were copper-dependent monooxygenases, 80% of which were from the AA9 family and the rest from the AA11 family.

Five genes encoding CFEM (common in fungal extracellular membrane) effector proteins were identified in the *V. inaequalis* secretome. The CFEM domain, a conserved eight-cysteine sequence, is found only in fungi, but not in all fungi, and has been implicated in multiple aspects of virulence [57]. The CFEM domain of Pth11, a key G protein-coupled receptor in *Magnaporthe oryzae*, has been reported to be required for pathogenesis, and the proper formation of appressorium and appressorium-like structures [58]. An *Alternaria alternata* AltA1-like effector was observed in the *V. inaequalis* secretome. AltA1 is implicated in fungal pathogenesis and has been reported to act as a virulence factor which blocks multiple plant defense mechanisms and promotes fungal infection [59]. Egh16-like virulence factors were also noticed. This family of effectors is involved in the appressorium formation and virulence of pathogens [60]. Effectors containing CYSTM domains were noted in the *V. inaequalis* secretome. This domain has been reported to play a role in stress response or resistance to stress, especially to toxicants [61,62]. Effectors belonging to the phospholipase 2 (PLA2) class were also noticed. Rafiei and coworkers [63] reported that the PLA2 effector of the fungal pathogen *Verticillium longisporum* targets the host nucleus and suppresses host PTI defense responses. Moreover, a necrosis-inducing effector protein (NPP1) was also identified in the *V. inaequalis* secretome. These proteins are widely distributed in different taxa of plant pathogens and act as virulence factors for necrotic or semi-biotrophic plant pathogens [64]. 

### 3.6. Expression Profiling of V. inaequalis Secretome

Studies of host–pathogen interactions help explain disease biology and develop preventive strategies. The studies mainly focus on secreted fungal proteins, particularly effectors secreted by host cells as they are often associated with compromised host defenses and the development of vulnerabilities through various mechanisms. In the present study, the RNAseq analysis revealed several differentially expressed genes (DEGs) required for the *V. inaequalis* infection of apple leaves. Subtilase, the best-known serine protease important for critical processes such as immunity, symbiosis and plant-programmed cell death, showed a 10.22-fold increase in expression levels. Studies have shown that pathogens secreting subtilisin inhibitors can indirectly prevent immune protease activation [65]. Cellulase (endo-beta-1,4-glucanase) genes degrade cellulose (one of the most abundant wall structural elements in most plant cell types) and other polysaccharides with 1,4-glycosidic bonds. It was observed as the second most overexpressed gene (9.98-fold). We found an 8.12-fold increase in tryptophan synthase expression. Plant immunity is activated by inhibiting tryptophan synthase, which causes cell death by increasing salicylic acid production [66]. We hypothesize that *V. inaequalis* might be increasing the tryptophan synthase expression to reduce the host immune response. Gene expression of the RsRlpA effector was also found to be upregulated 6.59-fold. RlpA-like proteins are protease inhibitors that promote virulence by suppressing hypersensitivity reactions [67]. Furthermore, an effector identical to AltA1 showed a 5.7-fold increase in expression. As previously mentioned, AltA1 helps fungi resist plant defense systems and promotes infection. Other annotated enzymes that were upregulated include exopolygalacturonase C, phosphoinositol-specific phospholipase C, multicopper oxidase, concanavalin A-like lectinase, subtilase, epoxide hydrolase, β-lactamase, BD-FAE (bifunctional feruloyl and acetyl xylan esterase acting on complex natural xylan), serine carboxypeptidase, xylanase, and Gfa1 (required for the synthesis of chitin metabolic precursors). Several hypothetical proteins were also downregulated by *V. inaequalis*. Annotated effectors such as cutinase/acetyl xylan esterase, laccase 3, phospholipase A2, TAP-like protein, amidohydrolase, hydrophobic surface binding protein A (HsbA), CHRD domain, domain of unknown function (DUF4360), and CFEM domain-containing effectors were all differentially expressed and down-regulated. In addition, several other peptidases and CAZymes were also downregulated (Figure 5). 

## 4. Conclusions

The present study analyzed the secretome architecture of *V. inaequalis*, a fungal pathogen responsible for apple scab disease. A comprehensive study of the *V. inaequalis* genome, transcriptome, and secretome has identified several extracellular enzyme systems present in this pathogen which should aid in the search for antifungal or plant disease-resistant ingredients active against this pathogen. This study sheds light on the diversity of plant cell-wall-degrading enzymes in this fungus that may be crucial for its interaction with host. The vast majority of proteins in the *V. inaequalis* secretome repertoire were hypothetical proteins. More research is needed to functionally characterize them, which will aid in the discovery of additional virulence determinants of this pathogen, and understanding their role in the infection process will undoubtedly provide new insight into the underappreciated intricacy of host colonization by this plant pathogen.

## Figures and Tables

**Figure 1 pathogens-12-00066-f001:**
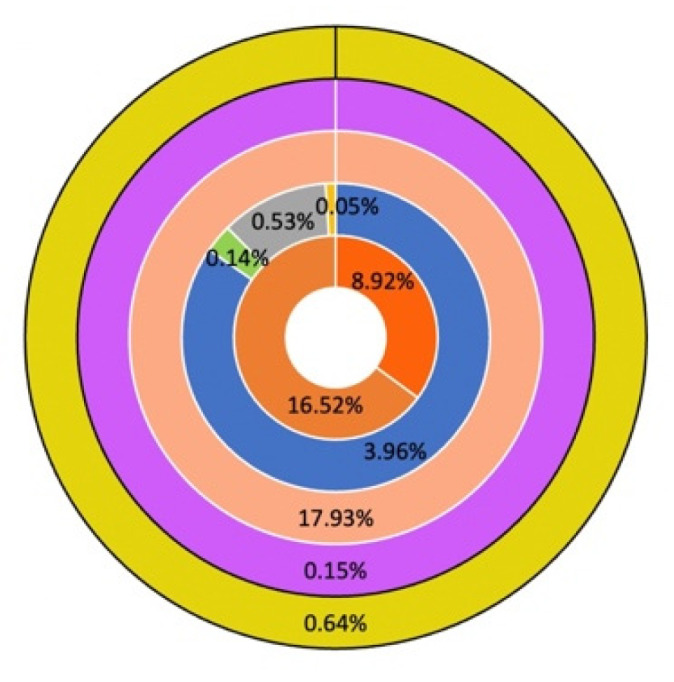
Distribution of transposable elements in the *V. inaequalis* genome. The repeats of the analyzed genomes are shown alongside (from the innermost ring outwards). Pie 1–5 represent LTR elements, DNA transposons, Unclassified, Small RNA and Satellites, respectively.

**Figure 2 pathogens-12-00066-f002:**
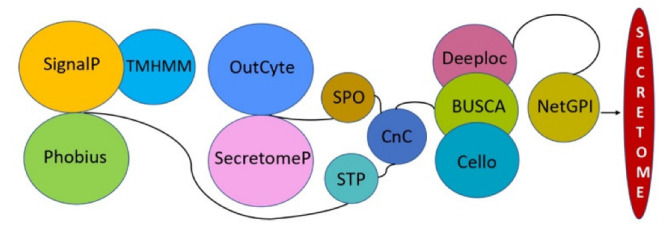
An *in silico* workflow used to dissect the secretome of *V. inaequalis*.

**Figure 3 pathogens-12-00066-f003:**
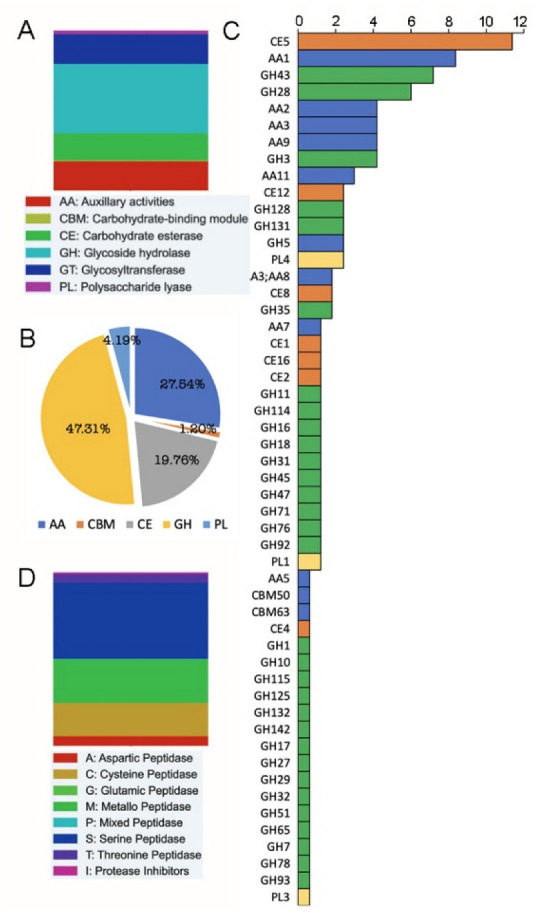
Relative abundance of CAZymes and peptidases of *V. inaquealis*. Distribution of the CAZyme family in the (**A**) genome and (**B**) secretome of *V. inaquealis*. (**C**) Frequencies of different classes of CAZymes in the *V. inaquealis* secretome. The bars indicate the number of CAZymes (**D**) Prevalence of the peptidase families in the *V. inaequalis* genome.

**Figure 4 pathogens-12-00066-f004:**
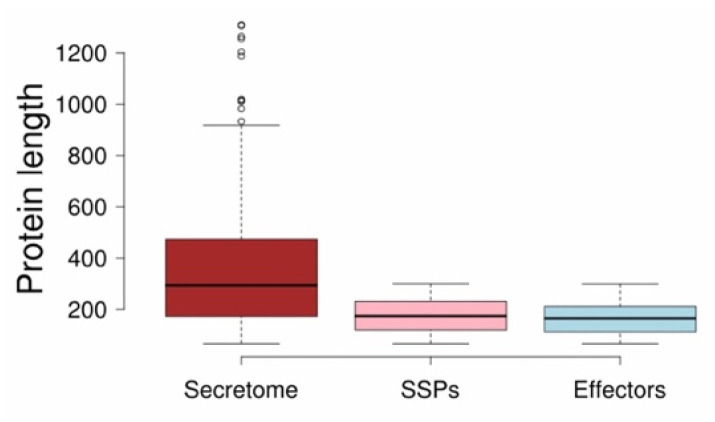
Variation in protein length across the secretome, small secreted proteins (SSPs), and effectome of *V. inaequalis*.

**Figure 5 pathogens-12-00066-f005:**
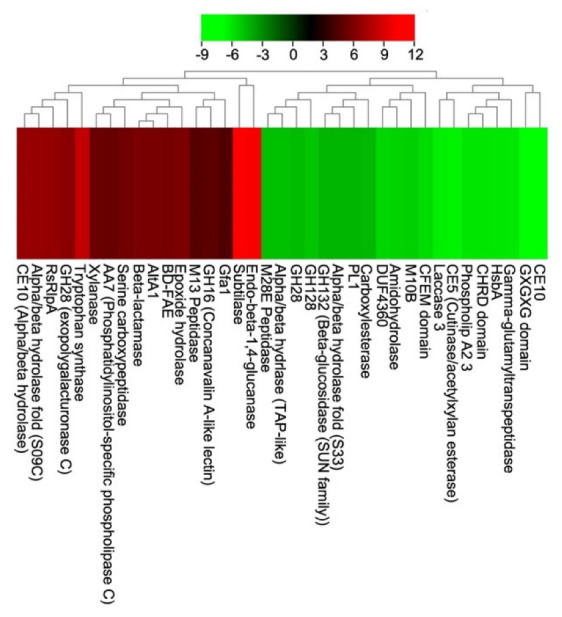
Gene expression profiles (logFC) of differentially expressed effector candidates from *V. inaequalis* during apple leaf invasion.

## Data Availability

The RNA-seq data reported in this study have been submitted to NCBI (BioProject ID: PRJNA909760) and other data files are available at Zenodo (doi: 7419358).

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
