# Peer review of "Secretomic Insights into the Pathophysiology of Venturia inaequalis: The Causative Agent of Scab, a Devastating Apple Tree Disease"

_pathogens, 2022, doi:10.3390/pathogens12010066_

Round 1

Reviewer 1 Report

In this paper, the authors identified the secreted proteins by using Transcriptome profiling combined with state-of-the-art bioinformatics analysis. They showed 328 proteins were classified as small secreted proteins (SSPs), with 76.52% of SSPs identified as anticipated effector proteins. The more prevalent CAZyme proteins were the enzymes engaged in plant cell wall disintegration (targeting pectin and xylanase), adhesion and penetration (Cutinases/acetyl xylan esterase), and reactive oxygen species formation (multicopper oxidases).

Comments:

1 The original data is needed to deposit this data in a public, community-supported repository.

2 There sentence is needed to improve:

“The genome of V. inaequalis has been ……….. 87 isolates from different populations of V. inaequalis.”

Author Response

Response to Reviewer1

Comments:

1 The original data is needed to deposit this data in a public, community-supported repository.

Response 1: As suggested by the reviewer, the original data has been submitted to the public, community-supported repositories; NCBI (BioProject ID: PRJNA909760) and Zenodo ((doi: 7419358). Thank you.

2 There sentence is needed to improve:

“The genome of V. inaequalis has been ……….. 87 isolates from different populations of V. inaequalis.”

Response 2: There was an unnoticed mistake owing to the reference citation formatting change, which created a confusing sentence. We have corrected it in the revised manuscript. Thank you.

Reviewer 2 Report

The authors worked on the deposited and published genome of Venturia inaequalis EU-B04. Using bioinformatics analyses they annotated, and classified genes into different categories depending on function. The main attention is put on the genes coding for potential pathogenicity factors. The idea of work and the methodology are generally very good, but the main drawback of this manuscript is the fact that it describes the results but it DOESN’T  SHOW them.

The Authors conclude in the paper that “The present study provides data that can be used to develop new strategies for controlling apple scab.” but the paper doesn’t show any details. When authors annotated the genome and identified the genes potentially involved in different pathways they should deposit annotated genome in the GenBank and provide the list of genes, their sequences, and identified functions as a supplementary table. The table will be big but supplementary material will be very welcome. Authors write about CAZyme proteins, Cutinases/acetyl xylan esterase, multicopper oxidases, or members of the S9 prolyl oligopeptidase family but the only result shown is that they exist in the genome which is not novel and could be expected. The novelty of the manuscript would be presenting what kind of genes are on the genome, and what function they have according to the results of the analysis performed by the Authors. In the present form, the paper will have only a small impact on the science and it will be not cited. However, the addition of the table with all genes description (which, I’m sure, the Authors have ), and the deposition of the annotated genome would definitely increase the impact of the paper.

Author Response

Response to Reviewer 2 Comments

Comments:

The Authors conclude in the paper that “The present study provides data that can be used to develop new strategies for controlling apple scab.” but the paper doesn’t show any details. When authors annotated the genome and identified the genes potentially involved in different pathways they should deposit annotated genome in the GenBank and provide the list of genes, their sequences, and identified functions as a supplementary table. The table will be big but supplementary material will be very welcome. Authors write about CAZyme proteins, Cutinases/acetyl xylan esterase, multicopper oxidases, or members of the S9 prolyl oligopeptidase family but the only result shown is that they exist in the genome which is not novel and could be expected. The novelty of the manuscript would be presenting what kind of genes are on the genome, and what function they have according to the results of the analysis performed by the Authors. In the present form, the paper will have only a small impact on the science and it will be not cited. However, the addition of the table with all genes description (which, I’m sure, the Authors have), and the deposition of the annotated genome would definitely increase the impact of the paper.

Response: 

Reviewer 2 expressed the same concerns about the availability of the generated data reported in this study, as noted by the Academic editor and reviewer 1. We took the advice into account and made all the data public (NCBI (BioProject ID: PRJNA909760) and Zenodo (doi: 7419358)). Thank you.

Round 2

Reviewer 2 Report

The required data files are now provided as supplementary files.